# Emotional arousal in 2D versus 3D virtual reality environments

**Feng Tian[1], Minlei Hua[1], Wenrui Zhang[1], Yingjie Li[2,3]\*, Xiaoli Yang[4]**

**1** Shanghai Film Academy, Shanghai University, Shanghai, China, **2** Shanghai Institute for Advanced Communication and Data Science, Shanghai University, Shanghai, China, **3** School of Communication and Information Engineering, Shanghai University, Shanghai, China, **4** Department of Electrical and Computer Engineering, Purdue University Northwest, Hammond, Indiana, United States of America

\* liyj@i.shu.edu.cn

**Data Availability Statement:** All relevant data are within the manuscript and its Supporting Information files.

## Abstract

Previous studies have suggested that virtual reality (VR) can elicit emotions in different visual modes using 2D or 3D headsets. However, the effects on emotional arousal by using these two visual modes have not been comprehensively investigated, and the underlying neural mechanisms are not yet clear. This paper presents a cognitive psychological experiment that was conducted to analyze how these two visual modes impact emotional arousal. Forty volunteers were recruited and were randomly assigned to two groups. They were asked to watch a series of positive, neutral and negative short VR videos in 2D and 3D. Multichannel electroencephalograms (EEG) and skin conductance responses (SCR) were recorded simultaneously during their participation. The results indicated that emotional stimulation was more intense in the 3D environment due to the improved perception of the environment; greater emotional arousal was generated; and higher beta (21–30 Hz) EEG power was identified in 3D than in 2D. We also found that both hemispheres were involved in stereo vision processing and that brain lateralization existed in the processing.

## 1 Introduction

In recent years, the integration of movies and VR technology has become an important breakthrough for traditional screen movies [1]. According to emotional gratification theory in media use, the desire to experience emotions is a key motivation for the use of entertainment media, especially in the form of movies [2]. VR movies attract people not only due to the advantages of omnidirectional stereoscopic images but also due to their significant emotional effects [3]. Numerous studies have provided evidence that VR is an effective method for eliciting emotions [4]. The VR technique is usually regarded as a better tool to provide stronger emotional arousal responses than other display techniques, such as ordinary 2D images [1].

VR technology can be used in many different forms, such as desktop VR, augmented VR, immersive VR and distributed VR (DVR). Macroscopic immersive VR (VR-2D) is one of the major techniques used in VR movies. It has spread widely throughout cyberspace because VR-2D has small bandwidth and low cost and is easy to shoot, produce and transfer. Stereoscopic environmental viewing conditions (VR-3D) can be obtained only through the use of

**Funding:** This work was supported by the National Social Science Fund of China under Grant No.17BC043.

**Competing interests:** The authors have declared that no competing interests exist.

professional stereoscopic VR camera shooting or by rendering production with an animation software engine, as shown in Fig 1. VR-3D can reproduce virtual environment videos in a realistic way, which allows users to have more realistic reactions than VR-2D. Recent studies have found that different virtual environment technologies (e.g., desktops, head-mounted displays, or fully immersive platforms) have different effects on emotional arousal [1].

According to Marin-Morales' literature review [5], immersive VR, which allows researchers to simulate environments under controlled laboratory conditions with high levels of sense of presence and interactivity, is becoming more popular in emotion research. This work highlights the evolution of immersive VR use and in particular, the use of head-mounted displays, in emotion recognition research in combination with implicit measures. However, little research has been conducted to determine the differences in emotional effects that are evoked by VR-2D and VR-3D.

Most research on desktop images has shown that 3D does not bring greater emotional arousal than 2D. Banos et al. [6] conducted comparative experiments on emotional responses in 2D and 3D environments, which indicated that there were no significant differences in subjective presence and emotional arousal between the two modes. Furthermore, some objective physiological measurements, such as electrodermal activity (EDA), have been used in research on the emotional effects of 2D or 3D displays in addition to subjective questionnaires. EDA is a sensitive psychophysiological index of the changes in autonomic sympathetic arousal that are integrated with emotional and cognitive states [7]. Bride et al. [8] studied the emotional impact of stereoscopic films and demonstrated that 3D films do not elicit stronger emotional reactions than 2D films with the same content by using both EDA measurements and self-reports. Peperkorn et al. [9] compared the fear emotions that were generated by spider images in 3D and 2D display modes and found that higher levels of subjective fear were evoked by 3D displays but that were no physiological differences in skin conductance level (SCL) of stereoscopy vs. monoscopy that were related to emotional arousal. It should be noted that all of the above research was conducted using ordinary desktop computers to display 2D or 3D images.

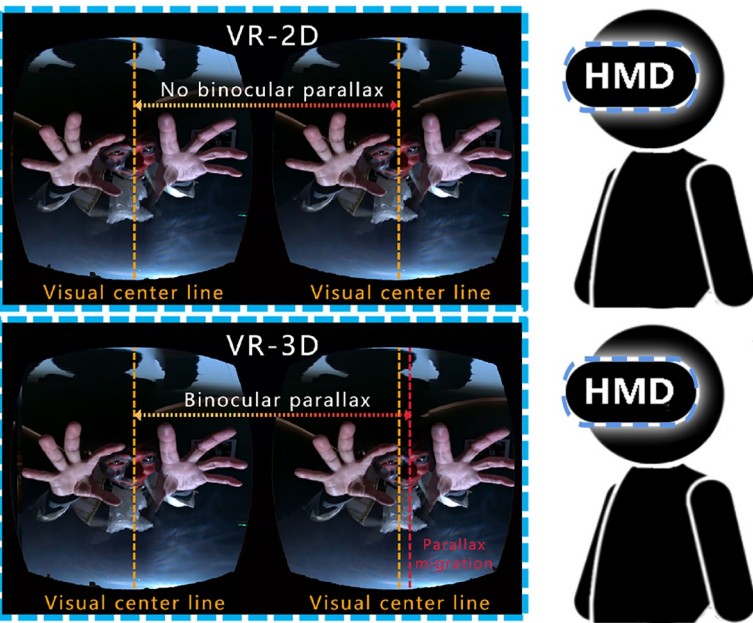

**Fig 1. Comparison of internal helmet images in VR-2D and VR-3D modes.**

Previous research has shown that the VR environment causes higher emotional arousal intensities compared to ordinary 2D displays with the same content. Most studies using subjective questionnaires have reported that user engagement in VR can be significantly increased and combined with more emotional involvement [10]. Kim et al. [11] found that, when compared with desktop platforms, both head-mounted displays (HMD) VR and six-wall fully immersive cave systems elicited concordant and greater changes in emotional arousal with discordant changes in emotional valence (i.e., HMD elicited more unpleasant effects than desktops). Higuera-Trujillo et al. [12] collected EDA and heart rate variability (HRV) data to compare 2D, 360˚ and VR. Their analysis showed that VR offers results that are closest to reality according to the physiological responses of the participants. These are important previous research efforts regarding comparisons of 2D and VR content. Moreover, Ding et al. [1] compared the emotional effects of cinematic VR (CVR) films (they did not clarify whether the VR films were 3D or 2D) and 2D films based on both self-reports and skin temperature analysis. The results showed more rapid and steadier skin temperature declines in the CVR group than in the 2D group. They concluded that the emotional effects of negative VR films were obviously stronger than those of 2D films.

In 2008, Hasson et al. [13] proposed the concept of neurocinematics, which demonstrated that the influence of images on the audience can be measured through brain activity. Recently, additional research has focused on the neural mechanisms underlying the VR environment, particularly by using electroencephalogram (EEG) analysis. Since EEGs provide a direct measure of brain activity on the scalp, EEG analysis has been used in 2D/3D studies for years. For example, by using EEG recordings, Slobounov et al. [14] demonstrated that the fully immersive 3D-enriched environment required allocation of more brain and sensory resources for cognitive/motor control than 2D presentations. Kober et al. [15] concluded that 3D provided experiences with more presence than 2D during virtual maze exploration tasks. However, these EEG studies are not directly related to emotion effects. To date, only limited studies have been conducted to investigate emotional effects in VR environments using EEG. He et al. [16] found that head-mounted VR environments made viewers perceive more reality than 2D environments, and more dimensions of sensory emotions and thinking were awakened. EEG analysis has unique advantages in mood detection; to the best of our knowledge, there are no published studies using EEG that explore the effects of stereo vision on emotional arousal [17]. Moreover, Tucker et al. [18] suggested that asymmetric brain activity patterns may be associated with specific emotional states. Since then, brain asymmetry based on EEG analysis has been widely investigated during the presentation of films that elicit emotional responses [19]. Balconi et al. [20] noted asymmetry in the comprehension of facial expressions of emotions and found that right and left side responses varied as a function of emotional types, with increased right frontal activity for negative, aversive emotions vs an increased left response for positive emotions. However, no studies have shown brain lateralization in VR or 3D stereoscopic environments.

In recent years, a combination of the characteristics of EEG data and machine learning algorithms to identify emotional states has become a new method. Marín-Morales et al. [21] performed an experiment that involved EEG and electrocardiography (ECG) recordings. They extracted features from these signals and fed them into a support vector machine (SVM) classifier to predict the arousal and valence perceptions of the subject. This is the first study that used EEG in an immersive scenario in combination with a machine learning algorithm to recognize emotional states. Later, in 2019, Marín-Morales et al. [22] developed an emotion recognition system using the SVM method combined with HRV and EEG data to identify arousal and valence. The literature showed that 3D is very valuable in applied research and analysis.

The main objective of this research is to compare the effects of VR-2D and VR-3D videos on the emotional arousal of participants by conducting multichannel physiological experiments by combining electroencephalography with skin electroencephalography. The corresponding research questions (RQs) are as follows:

RQ1: Is there a difference in the neural mechanism of emotional arousal that is stimulated by VR-2D or VR-3D?

RQ2: What EEG characteristics are significantly correlated with the intensity of emotional arousal?

RQ3: Which hemisphere of the human brain shows a greater effect on the perception of the stereoscopic environment?

## 2 Materials and methods

Several methods exist to measure emotional arousal, including self-reports, EEG and EDA [23]. In this paper, the subjective scale Positive and Negative Affect Schedule (PANAS) combined with Self-Assessment Manikin (SAM) was selected, and EEG and dermatoelectrical EDA were selected for objective physiological data collection.

### 2.1 The participants

In this experiment, 40 volunteers were recruited from universities in Shanghai and were randomly assigned to VR-2D or VR-3D groups equally, with 50% males and 50% females in each group. The average age of the subjects was 22.19±2.13 years, and the average years of education were 16.04±2.10 years. To exclude the learning transfer effect caused by the two experimental modes and avoid irritability or fatigue effects caused by the prolonged experimental time, each participant could only participate in one group of experiments, as was determined by the experimental grouping method of Kim et al. [24], Ding et al. [1], and He et al. [16]. All participants were tested between 9:00 AM-11:30 AM and 14:00 PM-16:30 PM to eliminate the potential confounding effects of circadian rhythms. The Self-rating Depression Scale (SDS) was completed before the experiment and included two items related to mental and emotional symptoms, eight items related to somatic disorders, two items related to psychomotor disorders, and eight items related to depressive psychological disorders. The scale is simple to use and can directly reflect the subjective feelings of depressed patients. The experiment was approved by the Shanghai Clinical Medicine Ethics Committee.

### 2.2 Materials and tasks

All VR stimuli in this experiment were produced by the Shanghai Film Academy of Shanghai University. The city promotional film was used as the positive emotional stimulation (provided by Motion Magic, Inc), videos of natural situations were used as the neutral emotional stimuli, and videos of the horror corridor were used as the negative emotional stimuli, as shown in Fig 2. The positive, neutral and negative VR video materials contain 20 different contents. The image lengths were 4 seconds, the resolution was 4096*2048, and the frame speed was 30 frames per second. We used Nokia's OZO VR video camera to shoot the VR-3D videos in this experiment and then used Adobe's Premiere video production software to process the VR-3D materials. VR-2D was obtained by eliminating the parallax of the left and right eye images in VR-3D.

An additional 15 volunteers (mean age 21.56 ± 2.67 years old) were recruited to rate the valence and arousal of the materials before they were used in the experiment. On a scale of 1 to

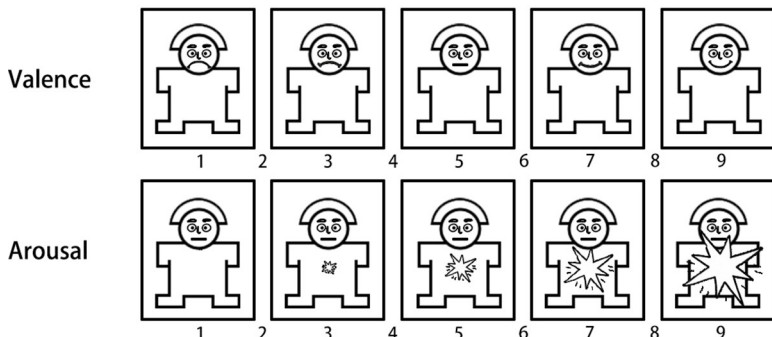

**Fig 2. The experimental materials were divided into the VR-2D group and VR-3D group.** The positive, neutral and negative VR video materials have 20 different contents.

9, 9 represents the highest level of pleasure and the highest level of arousal, as shown in Fig 3. The averages of the valence and arousal values in SAM are shown in Table 1. The results showed that the three types of materials were significantly different from each other and met the experimental requirements.

### 2.3 Procedures

The experimental paradigm of Decety et al. [25] was adopted for use in this study. As shown in Fig 4, there were 2 blocks in the experiment. Each block contained 30 trials, which were defined as 20 positive trials and 10 neutral trials or 20 negative trials and 10 neutral trials, and the trial order was random. Both types of blocks appeared twice in random order. Each trial displayed a black screen with a white cross, which was followed by a 4-second image. The subjects were required to pay attention to the video. After the image disappeared, the subjective rating of the image was entered on a keyboard. The subjective score adopted the SAM scale, which was used to determine whether the participants were fully engaged in the experiment and to provide them with rest time. The program was written using Unity2018a (Unity Technologies, San Francisco, USA).

### 2.4 Data recording and processing

EEGs were recorded with an EEG system from Neuracle (Neuracle Technology, Changzhou, China). EEG recordings were obtained with standard Ag/AgCl 64-channel scalp electrodes, referenced to the Cpz, and an AFz ground was used. EEGs were continuously recorded at a sampling frequency of 1000 Hz and were bandpass filtered from 0.5 Hz to 30 Hz, and the inter-electrode impedances were kept below 5 kΩ. The skin electricity acquisition device was a

**Fig 3. Schematic diagram of the SAM (Self-Assessment Manikin) scale.** The Self-Assessment Manikin (SAM) was used to rate the affective dimensions of valence and arousal.

**Table 1. The average value (mean ± S.D.) of valence and arousal in SAM (Additional 15 volunteers).**

|  | Positive | Neutral | Negative |
|---|---|---|---|
| Arousal | 4.63±1.26 | 2.99±1.62 | 5.56±1.46 |
| Valence | 6.47±0.74 | 5.26±0.48 | 2.77±0.68 |

BIOPAC MP150 system (BIOPAC Systems Inc., Goleta, USA), and the VR headset device was an HTC Vive (HTC Corporation, Taoyuan City, Taiwan). Fig 5 shows the experimental process and equipment connections.

The EEG and EDA data were collected simultaneously while participants viewed VR-2D and VR-3D videos.

EDA data preprocessing includes reducing the sampling frequency, finding the mean value and aligning the data. EDA data can be directly exported into MATLAB after data sampling. EDA data management and extraction used Ledalab, which is free software developed by Benedek et al. [26], for data optimization processing. In the data optimization process, data that are preset within the response window and meet the minimum amplitude criteria are determined to be peak SCRs that are caused by the stimuli. The flowchart for the signal processing used in this study is shown in Fig 6.

All collected EEG data were preprocessed in offline mode using the EEGLAB toolbox, including baseline adjustments and artifact removal from the EEGs and HMDs. The EEG signal processing flowchart for this study is shown in Fig 7.

For the problem of signal generation on some channels, we interpolated the data segment of the channel, that is, we replaced the signal value of the bad channel with data from 2–3 channels adjacent to the channel as a star interpolation.

Data from total of twenty VR-2D and VR-3D subjects were collected. In the process of brain electrical signal preprocessing, the SDS scores of one subject in each of the VR-2D and VR-3D groups did not meet the experimental requirements. Therefore, the final data used in this experiment were from 19 people in the VR-2D group and 19 people in the VR-3D group. During the process of EEG data collection, the subjects were required to remain calm to prevent signal jamming caused by VR headsets.

**2.4.1 PANAS and SAM.** The Self-Assessment Manikin (SAM) is a nonverbal image-oriented assessment scale that was designed by Bradley and Lang [27], which can directly measure human emotional responses to various stimuli, including valence and arousal. The Positive and Negative Affect Schedule (PANAS) was developed by Watson et al. [28]. It contains 20 items that are divided into two independent dimensions, namely, positive affect (PA) and negative affect (NA), with 10 items in each dimension. Both SAM and PANAS are based on the subjective assessments of participants.

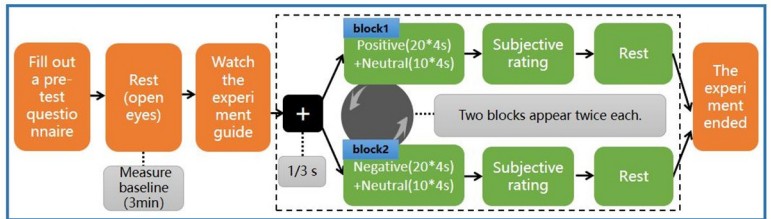

**Fig 4. Flowchart of the cognitive psychological experiment.** The experiment consists of the emotion task and subjective rating. There were 2 blocks in the experiment. Each block contained 30 trials, and the trial order was random. After the image disappeared, the subjective rating of the image was entered via the keyboard.

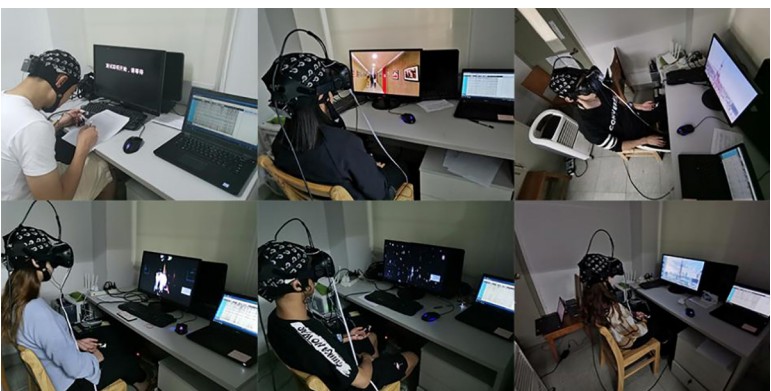

**Fig 5. Experimental process and equipment connections.**

In the PANAS subjective behavior scale, the postviewing emotional measurement results were subtracted from the baseline before viewing to obtain the change values before and after viewing. One-way analysis of variance (ANOVA) was used in SPSS to compare positive and negative films. In the SAM subjective behavior scale, the results of emotional valence and arousal degree of subjects were obtained by using two-factor repeated measurement analysis of variance in SPSS.

**2.4.2 EEG power.** The artifact free data were bandpass filtered in EEGLAB (based on different frequency bands, the thresholds of high-pass and low-pass filters were selected), and alpha band (8–13 Hz), beta band (13–30 hz) which was divided into three subbands: β1 (13–18 hz), β2 (18–21 hz), and β3 (21–30 hz) were obtained for further analysis.

As shown in Eq 1, the sum of the squares of all points in the frequency band represented the power of a data segment,

$$E(k) = \frac{1}{n}\sum\nolimits_{i=1}^{n} x(k)_i^2 \tag{1}$$

where $k$ represents the number of trials in the data segment, $n$ represents the number of data points in each segment, and $x(k)_i$ represents the value of the ith point in the kth data segment.

**2.4.3 SCR.** EDA incorporates both slow shifts in basal skin conductance level (SCL) and more rapid transient events, that is, skin conductance responses (SCRs), which have also been referred to as galvanic skin responses (GSR) [7]. For example, Ravaja et al. [23] used the skin conductance response (SCR) to view the subjective scale of photographs and showed that the SCR was positively correlated with arousal intensity. In this study, each stimulating video clip was displayed for approximately 4 seconds, which is a short-term event. We therefore chose SCRs for analysis. The number of these "peaks" is represented as numSCRs, and the mean power of the peaks is represented as avgSCRs.

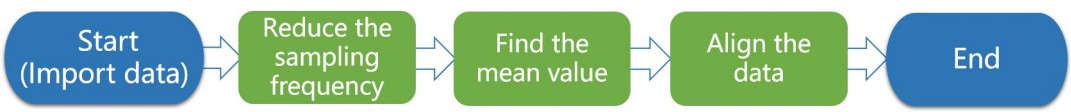

**Fig 6. The flowchart of EDA data processing.**

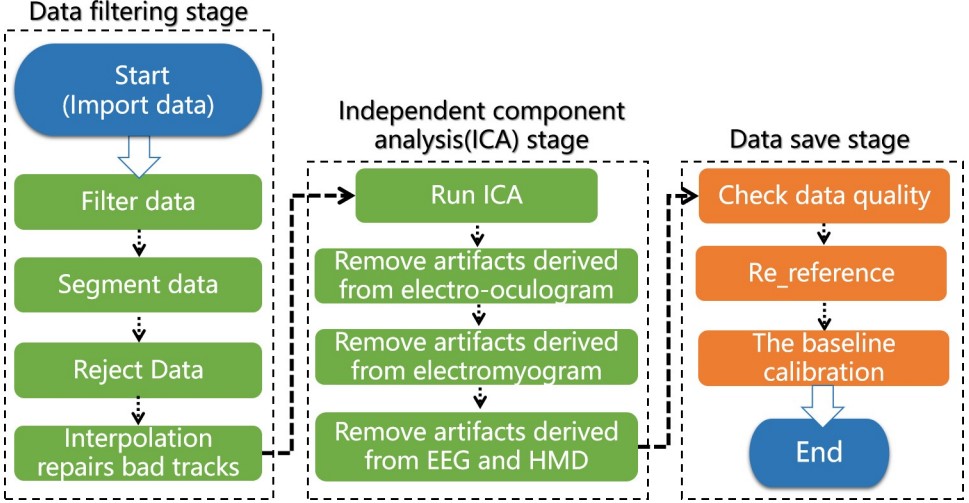

**Fig 7. The flowchart of EEG data processing.**

## 2.5 Statistical analysis

ANOVA was used with two within-group factors of emotion (e.g., positive, neutral and negative); region (e.g., frontal, parietal, central, temporal and occipital) and one between-group factor of group (3D and 2D) for EEG power.

Considering that the sample scale was not large, we repeated the ANOVA to investigate the hemisphere effect with two within-group factors of emotion (e.g., positive, neutral and negative); hemisphere (e.g., left and right) and one between-group factor of group (e.g., 3D and 2D) for EEG power. Simple effects analysis was performed if any interaction among factors was found. For the EEG data, we selected five groups of electrodes that represent five brain regions: frontal region (Fz, F3, F4, FCz, FC3, and FC4); parietal region (Pz, P3, and P4); central region (Cz, C3, C4, CP3, and CP4); temporal region (TP7, TP8, T7, T8, P7, and P8); and occipital region (POz, PO3, PO4, Oz, PO7, and PO8), as shown in Fig 8.

For the EDA data, the mean amount and energy of SCR excitation were calculated by means of statistical averages. ANOVA was used with one within-group factor of emotion (e.g., positive, neutral and negative) and one between-group factor of group (e.g., 3D and 2D) for avgSCRs.

All analyses were conducted at a 0.05 level of significance. Multiple comparisons were corrected with the Bonferroni correction. All statistical analyses were conducted using SPSS 22.0.

## 3 Results

### 3.1 Behavioral data

The valence and arousal data using the SAM scale are shown in Table 2. The main effect of emotional valence on the SAM scale revealed that positive valence was significantly higher than that of the neutral material, and that negative valence was significantly lower than that of the neutral material. The main effect of emotional arousal revealed that negative arousal was higher than positive arousal, and positive arousal was higher than neutral arousal. Overall, the evaluation results for emotional valence and arousal showed that the subjects could clearly distinguish the emotional significance of the images used in this study.

The group effect between VR-2D and VR-3D modes on the emotional dimensions (e.g., valence and arousal) was not significant ($F(2, 72) = 1.864$, $p = 0.162$, $\eta^2 = 0.049$). In other

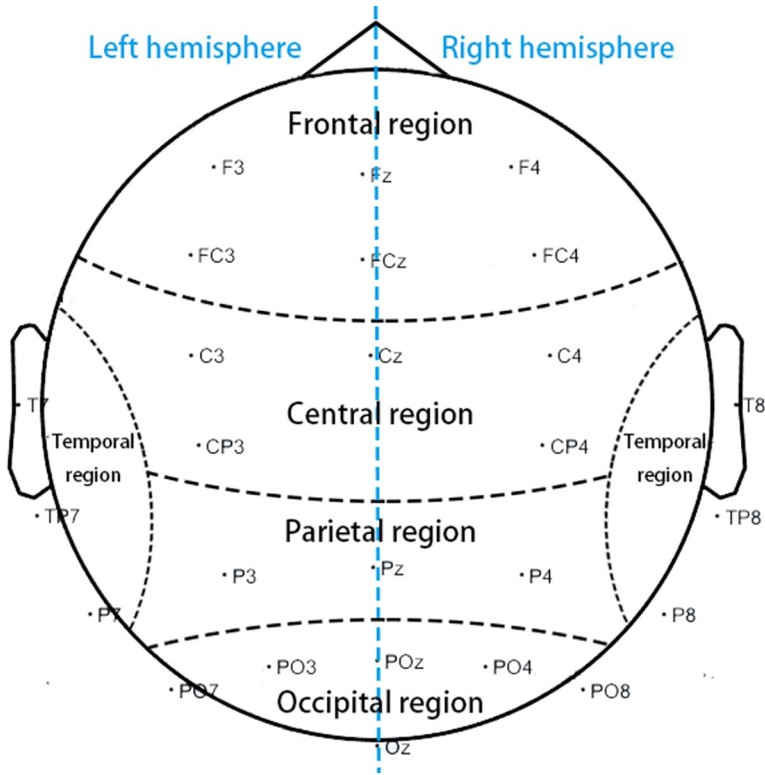

**Fig 8. Layout of the locations of the EEG scalp electrodes on brain regions.** Five groups of EEG scalp electrodes representing five brain regions: frontal, parietal, central, temporal, and occipital.

words, participants did not feel emotional differences from valence and arousal between the two display modes.

## 3.2 EDA data analysis

As shown in Table 3, the results for avgSCRs showed a primary effect of emotion ($F(2,72)$ = 2.373, p = 0.000, $\eta^2$ = 0.361), which revealed that negative emotion was significantly greater than positive emotion and that positive emotion was significantly greater than neutral emotion. The numSCRs of negative videos were higher than those of positive videos, and the numSCRs of positive videos were higher than those of neutral videos, both in VR-2D and in VR-3D. In other words, the SCRs that were induced by the "same" content in VR-3D were higher than those in the VR-2D viewing environment.

## 3.3 EEG analysis

### 3.3.1 Emotion effect.

**Table 2. The average value (mean ± S.D.) of valence and arousal in SAM (38 volunteers).**

|  |  | Positive | Neutral | Negative |
|---|---|---|---|---|
| VR-2D | Arousal | 4.33±1.58 | 2.43±1.23 | 5.17±1.37 |
|  | Valence | 6.52±0.63 | 5.24±0.57 | 2.75±0.81 |
| VR-3D | Arousal | 4.61±1.71 | 3.26±1.49 | 5.79±1.19 |
|  | Valence | 6.32±0.73 | 5.20±0.31 | 2.71±0.77 |

**Table 3. The avgSCRs (mean ± S.D.) and numSCRs under different movie-watching modes.**

| | VR-2D | | VR-3D | |
|---|---|---|---|---|
| | **AvgSCRs** | **NumSCRs** | **AvgSCRs** | **NumSCRs** |
| Neutral | 0.41±0.15 | 145 | 0.43±0.18 | 165 |
| Positive | 0.60±0.23 | 147 | 0.72±0.32 | 174 |
| Negative | 0.87±0.37 | 218 | 0.93±0.39 | 257 |

- **α band**

The results for the α band showed a main effect of emotion ($F_{(2,72)}$ = .285, p = 0.003, $\eta^2$ = 0.149), which revealed that positive emotions elicited significantly greater alpha activity than negative emotions and greater alpha activity than neutral emotions. A significant main effect of region ($F_{(4,144)}$ = 32.701, p = 0.000, $\eta^2$ = 0.476) revealed that the power in the occipital region was larger than that in other regions. We also found an emotion*group interaction effect at the trend level on accuracy ($F_{(2,72)}$ = 2.775, p = 0.069, $\eta^2$ = 0.072).

To investigate the interaction of emotion*group, ANOVA with a within-factor of emotion was performed for each group. The analysis showed that the emotion effect in the VR-3D group ($F_{(2,36)}$ = 6.227, p = 0.005, $\eta^2$ = 0.257) was large and that positive emotions (P = 0.03<0.05) were significantly higher than neutral and negative emotions, as shown in Fig 9. No emotion effect was found in VR-2D group.

In the follow-up analysis, ANOVA with a within-factor of group was performed for each emotion. The analysis showed that VR-3D elicited greater alpha activity for positive emotions ($F_{(1,36)}$ = 3.516, p = 0.069, η2 = 0.089) than VR-2D.

Our results demonstrated that VR-3D elicited greater beta activity than VR-2D, as shown in Fig 10.

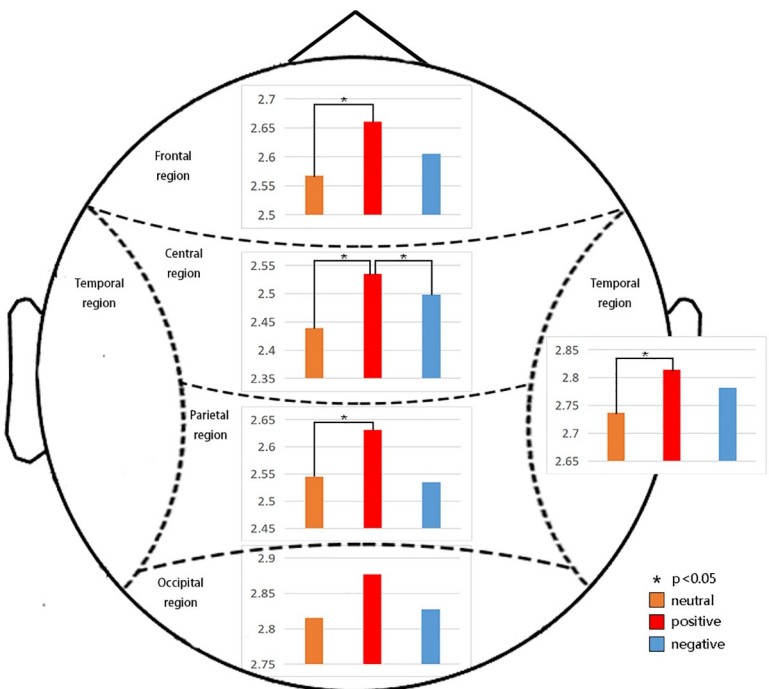

**Fig 9. Significant differences in different brain regions under VR-2D or VR-3D modes for the α band.** The emotion effect in the VR-3D group was large, and positive emotions (P = 0.03<0.05) were significantly higher than neutral and negative emotions.

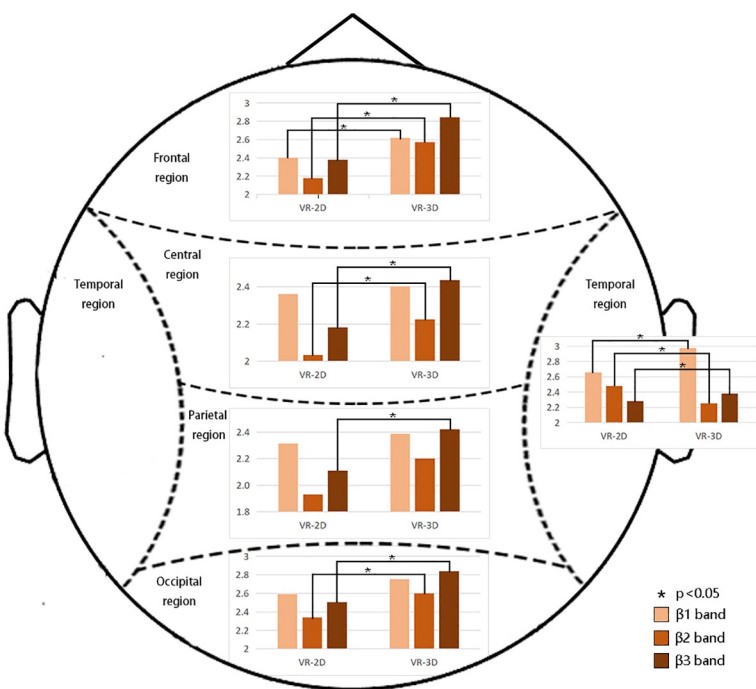

**Fig 10. Significant differences in different brain regions under VR-2D or VR-3D modes for the β1, β2 and β3 wave bands.** The results demonstrated that VR-3D elicited greater beta activity than VR-2D.

- **β1 band**

The results for the β1 band showed a main effect of emotion ($F(2,72) = 4.067$, $p = 0.021$, $\eta^2 = 0.101$), which revealed that neutral and negative stimuli elicited greater beta activity than positive stimuli. A significant main effect of region ($F(4,144) = 65.776$, $p = 0.000$, $\eta^2 = 0.646$) revealed that the power in the temporal region was larger than that in other regions. We found significant interaction effects of emotion*region ($F(8,288) = 2.258$, $p = 0.024$, $\eta^2 = 0.059$) and region*group ($F(4,144) = 5.283$, $p = 0.001$, $\eta^2 = 0.128$). To investigate the interaction of emotion*region, ANOVA with a within-factor of emotion was performed for each region. In the frontal region, the emotion effect in the positive stimulus group ($P = 0.036 < 0.05$) was significantly higher than that in the neutral stimulus group. In the parietal region, the emotion effect for the negative stimuli ($P = 0.015 < 0.05$) was significantly higher than that for the neutral stimuli. In the temporal region, the emotion effect for negative stimulation ($P = 0.044 < 0.05$) was significantly higher than that for neutral stimulation. In the occipital region, the emotion effect for positive stimuli ($P = 0.007 < 0.05$) was significantly higher than that for neutral stimuli. To investigate the interaction of region*group, ANOVA with a within-factor of group was performed for each region. This analysis showed that VR-3D elicited greater beta activity than VR-2D in the frontal region ($F(1,36) = 5.284$, $p = 0.027$, $\eta^2 = 0.128$) and temporal region ($F(1,36) = 7.391$, $p = 0.001$, $\eta^2 = 0.170$).

- **β2 band**

The results for the β2 band showed a main effect of region ($F(4,144) = 105.68$, $p = 0.000$, $\eta^2 = 0.746$) and revealed that the power of the temporal region was also larger than that in other regions. Significant interaction effects of emotion*region ($F(8,288) = 4.802$, $p = 0.000$, $\eta^2 = 0.118$) and region*group ($F(4,144) = 3.428$, $p = 0.010$, $\eta^2 = 0.087$) were found. To investigate

the interaction of emotion*region, ANOVA with a within-factor of emotion was performed for each region. In the temporal region, the energy induced by negative stimulation (P = 0.018<0.05) was significantly higher than that induced by positive stimulation. In the occipital region, the energy induced by negative stimuli (P = 0.047<0.05) and positive stimuli (P = 0.033<0.05) was significantly higher than that induced by neutral stimuli. To investigate the interaction of region*group, ANOVA with a within-factor of group was performed for each region. A significant group effect in the frontal region (F(1,36) = 17.278, p = 0.000, $\eta^2$ = 0.324); central region (F(1,36) = 9.213, p = 0.004; $\eta^2$ = 0.204), temporal region (F(1,36) = 10.292, p = 0.003, $\eta^2$ = 0.222); and occipital region (F(1,36) = 6.913, p = 0.013, $\eta^2$ = 0.161) was found. This analysis showed that VR-3D elicited greater β2 activity than VR-2D (see also in Fig 10). No group effect was found in the parietal region.

- **β3 band**

There was also a main effect of region (F(4,144) = 95.683, p = 0.000, $\eta^2$ = 0.727) in the β3 band. The interaction effects of emotion*region (F(8,288) = 5.041, p = 0.000, $\eta^2$ = 0.123) and region*group (F(4,144) = 3.201, p = 0.015, $\eta^2$ = 0.082) were significant. To investigate the interaction of emotion*region, ANOVA with a within-factor of emotion was performed for each region. However, no emotion effect was found in the frontal, parietal, central, temporal or occipital regions. To investigate the interaction of region*group, ANOVA with a within-factor of group was performed for each region. Group effects in the frontal region (F(1,36) = 24.242, p = 0.000, $\eta^2$ = 0.402); parietal region (F(1,36) = 7.500, p = 0.010, $\eta^2$ = 0.172); central region (F(1,36) = 11.334, p = 0.002, $\eta^2$ = 0.239); temporal region (F(1,36) = 13.944, p = 0.001, $\eta^2$ = 0.279) and occipital region (F(1,36) = 10.814, p = 0.002, $\eta^2$ = 0.231) were found, which indicated that VR-3D elicited greater beta activity than VR-2D (see also in Fig 10).

In addition, it is interesting to note that increased frequency increased the number of regions showing group differences, i.e., from β1 to β3.

**3.3.2 Hemisphere effect.**   In the β band, the main effect of hemisphere (F(1,36) = 13.323, p = 0.001, $\eta^2$ = 0.269) showed that the beta activity in the left hemisphere was larger than that in the right hemisphere.

The results for the β band showed a significant interaction effect of the hemisphere*group effect (F(4,144) = 5.210, p = 0.028, $\eta^2$ = 0.126). To investigate the interaction of hemisphere*group, ANOVA with a within-factor of hemisphere was performed for each group. This analysis showed that the left hemisphere evoked significantly higher cortical activity than the right hemisphere in the VR-3D group (F(1,18) = 10.976, p = 0.004, $\eta^2$ = 0.379). No hemisphere effect was found in the VR-2D group.

ANOVA with a within-factor of group was also performed for each hemisphere and showed that VR-3D elicited significantly greater beta activity than VR-2D in both the left hemisphere (F(1,36) = 5.826, p = 0.021, $\eta^2$ = 0.139) and right hemisphere (F(1,36) = 5.144, p = 0.029, $\eta^2$ = 0.125).

## 4 Discussion

### 4.1 VR-3D elicits higher emotional arousal in EDA

Previous studies [6, 8, 9] have shown that 3D films do not enhance emotional benefits compared to 2D films with the same content. In our results from the SAM questionnaire, only differences among emotions were found. It should be noted that we could not use the PANAS questionnaire, as the positive and negative clips were interrupted by the neutral clips due to the use of videos with 4 s periods in this study. In another experiment, we invited two different groups of people to watch two types of continuous positive and negative videos (not clips),

**Table 4. Behavior score of PANAS emotion scale (mean).**

| Positive videos | | | Negative videos | | |
|---|---|---|---|---|---|
| Words | VR-2D | VR-3D | Words | VR-2D | VR-3D |
| enthusiastic* | 0.043 | 0.591 | distressed | 0.381 | 0.524 |
| interested* | -0.13 | 0.429 | upset | 0.143 | 0.455 |
| Determined | -0.227 | 0.19 | hostile | 0.091 | 0.524 |
| Excited | 0.087 | 0.667 | irritable | 0.191 | 0.273 |
| Inspired | 0.348 | 0.619 | scared* | 0.800 | 1.818 |
| Alert | -0.043 | -0.222 | afraid* | 0.545 | 1.682 |
| active* | -0.227 | 0.762 | ashamed | -0.217 | -0.045 |
| Strong | -0.318 | 0.158 | Guilty | -0.043 | 0.091 |
| Proud | 0.455 | 0.200 | nervous* | 0.217 | 1.182 |
| attentive | -0.318 | 0.143 | jittery* | 0.429 | 1.286 |

Note

* means p < 0.05.

VR-3D and VR-2D, respectively, and the contents were identical to the materials used in this study. The participants were asked to complete the PANAS scale prior to and after the experiment. The results are shown in Table 4 and Fig 11. VR-3D elicited greater emotional arousal than VR-2D regardless of the presence of positive or negative stimuli. The PANAS results confirmed that stereoscopic vision significantly increases arousal more than monoscopic vision. In other words, 3D causes increased emotional arousal compared with 2D in the VR environment. Moreover, the avgSCRs in VR-3D are higher than those in VR-2D, and the numSCRs generated in VR-3D are also greater than those in VR-2D in a unit time. It has been documented that SCR results are positively correlated with emotional arousal; therefore, we concluded that a VR environment with stereo vision can enhance emotional arousal compared with the corresponding 2D mode [3].

### 4.2 Mode differences between VR-3D and VR-2D reflected in EEG

The differences in image viewing modes had a significant effect on brain activity. Our results showed that VR-3D evoked significantly larger β3 band EEG activity than VR-2D in five brain regions. Li et al. [29] reported on brain fatigue and demonstrated that the power of EEG after

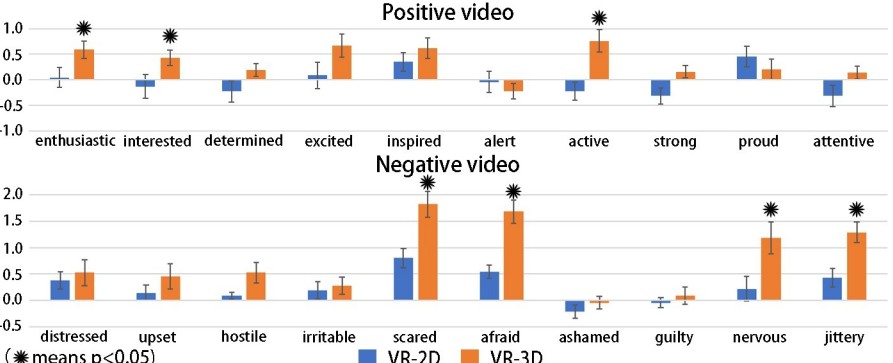

**Fig 11. Histogram of the PANAS emotion scale.** The VR-3D elicited greater emotional arousal than the VR-2D regardless of positive or negative stimuli. These words, enthusiastic, interested and active in the positive videos and scared, afraid, nervous and jittery in the negative videos showed significant differences.

viewing 3D images with glasses wearing was stronger than that under 2D conditions, which mainly occurred for waves at frequencies higher than 12 Hz. Tarrant et al. [30] found that VR intervention uniquely effected beta activity. Kweon et al. [31] found higher beta activity when subjects were exposed to VR videos than to desktop 2D for genre films. VR panoramic pictures are larger than desktop 2D, and the equipment used to display VR and 2D is different. Considering that the desktop 3D experiment requires wearing 3D glasses, it is difficult to confirm whether the difference between 3D and 2D glasses that they found was caused by stereo vision or fatigue caused by the device itself. Our research compares VR-3D and VR-2D when using the same VR helmet. Therefore, we infer that the higher beta activity evoked by VR-3D came from stereo vision processing. In Higuera-Trujillo's literature [12], the VR used for comparison is a VR-3D scene generated by a game engine, and 360° is the actual shot VR-2D mode. However, for this article, VR-3D and VR-2D were both obtained by shooting real scenes. Compared with game engine rendering, there are still visual differences in the details and realism of the pictures. Our conclusion also further confirms the results of Higuera-Trujillo, that is, VR-3D can produce greater emotional arousal, which may be caused by the stronger sense of reality of stereo vision.

In addition, the alpha band results indicated that alpha EEG is not sensitive to stereo vision processing since no group effect was found in the alpha band.

## 4.3 Brain lateralization in VR-3D and VR-2D

This study also clarified the nature of hemispheric effects evoked by VR video. Our results demonstrate that VR-3D increased beta power, especially in the left hemisphere. Previous studies have shown that the brain has a lateralization of emotional information processing. Balconi et al. [20] noted that the right- and left-side responses varied as a function of emotional type, with an increased left-side response for positive emotions. Hagemann et al. [32] proposed that the right hemisphere may play an important role in the automatic generation of emotional responses, whereas the left hemisphere may be involved in the control and modulation of emotional reactions. However, it is still unclear how brain lateralization acts on emotional arousal under VR conditions. In this study, we did not find an effect of lateralization of emotional VR stimuli. We found that the cortical activity of the left hemisphere in the β band was significantly higher than that of the right hemisphere only for VR-3D, but not for VR-2D. Therefore, we speculate that stereo vision processing is reflected in the increased energy of cortical activity of the left hemisphere. To date, there are no published studies that have examined EEG power in hemispheres associated with VR used in the context of an emotion processing paradigm; thus, it is difficult to directly relate our results to any previous reports.

When the brain is active, the neural activities of different regions are integrated at various spatial and temporal scales; this is termed the brain synchronization phenomenon. Li et al. [33] used the spectral features of the brain network to extract its synchronization features and proposed a method based on the fusion of traditional features and spectral features to achieve an adjustment of patient brain network synchronization abilities. By studying the ability of brain networks to synchronize, we can focus on describing the states and differences in the interactions between the two brain regions. Thus, in another recent study by our group [34], we compared neutral video in VR-2D and VR-3D through brain synchronization analysis. By constructing multi-band functional brain networks, we observed higher global efficiency in β3 (21–30 Hz) networks in the VR-3D group. This strongly proves the conclusion of this paper.

The limitations of this paper include the following: (1) The number of subjects participating in the experiment is not sufficiently large, which limits our analysis of the data. In the future, if EEG information can be classified and recognized with machine learning, more subjects are

needed, and more data are collected. (2) The artifacts between the HMD and EEG helmets easily effect data quality. To better handle data artifacts, the data should be processed in a more scientific way to eliminate the subjectivity of manual data processing.

## 5 Conclusion

The emotional responses and awakening viewer rhythms are the most important contents for film content creators and provide strong evidence to predict the market and public praise. The emergence of neuroaesthetics provides a scientific approach for the study of cinematography that is based on the acquisition of physiological data. The empirical research approach of VR video visual art theory can be conducted from the in-depth study of aesthetic cognition and neural mechanisms. In this study, we compared and evaluated the effects of two different modes, e.g., VR-2D and VR-3D, on emotional arousal by analyzing subjective and objective data that were obtained from cognitive psychological experiments. The results showed that the emotional stimulus in the stereo vision environment would affect people's perception and presence of the virtual reality environment to a greater extent, which would thereby generate greater emotional arousal. The stereo effect is mainly manifested in high-frequency brain activity, i.e., the β3 (21–30 Hz) band, which leads to brain asymmetry with a more active left hemisphere.

## Supporting information

**S1 Dataset. Minimal data set.**
(ZIP)

## Author Contributions

**Conceptualization:** Feng Tian.

**Data curation:** Feng Tian, Minlei Hua, Wenrui Zhang, Yingjie Li.

**Formal analysis:** Feng Tian, Yingjie Li.

**Funding acquisition:** Feng Tian.

**Investigation:** Feng Tian, Wenrui Zhang.

**Methodology:** Feng Tian, Yingjie Li.

**Project administration:** Feng Tian, Minlei Hua, Wenrui Zhang.

**Resources:** Feng Tian, Minlei Hua, Yingjie Li.

**Software:** Feng Tian.

**Supervision:** Minlei Hua.

**Visualization:** Minlei Hua, Wenrui Zhang, Yingjie Li.

**Writing – original draft:** Minlei Hua, Yingjie Li, Xiaoli Yang.

**Writing – review & editing:** Minlei Hua, Yingjie Li, Xiaoli Yang.

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
