## [Decision Letter · Decision Letter 0]

12 Apr 2021

PONE-D-20-38463

Emotional arousal in 2D versus 3D virtual reality environment

PLOS ONE

Dear Dr. Li,

Thank you for submitting your manuscript to PLOS ONE. After careful consideration, we feel that it has merit but does not fully meet PLOS ONE’s publication criteria as it currently stands. Therefore, we invite you to submit a revised version of the manuscript that addresses the points raised during the review process.

We look forward to receiving your revised manuscript.

Kind regards,

Christos Papadelis, Ph.D.

Academic Editor

PLOS ONE

Journal Requirements:

3. Please include your tables as part of your main manuscript and remove the individual files. Please note that supplementary tables should be uploaded as separate "supporting information" files.

4. We note that Figure s 1 and 3 includes an image of a participant in the study. 

Reviewers' comments:

Reviewer's Responses to Questions

**Comments to the Author**

1. Is the manuscript technically sound, and do the data support the conclusions?

Reviewer #1: Partly

2. Has the statistical analysis been performed appropriately and rigorously? 

Reviewer #1: No

3. Have the authors made all data underlying the findings in their manuscript fully available?

Reviewer #1: No

4. Is the manuscript presented in an intelligible fashion and written in standard English?

Reviewer #1: No

5. Review Comments to the Author

Reviewer #1: The paper present a timely analysis of the differences between 2d and 3d VR. It presents interesting insight for the scientific community. However, I have major concerns in particular with previous research and data processing.

Major concerns:

- The introduction is quite limited and the state of the art have not been properly shown. Many references need to be considered

Marín-Morales, J., Llinares, C., Guixeres, J., & Alcañiz, M. (2020). Emotion recognition in immersive virtual reality: From statistics to affective computing. Sensors, 20(18), 5163.

Emotion recognition review using immersive VR, which summarize all the research performed in the emotion field with VR, in particular comparing 360º and 3D stimuli, which is quite related to RQ3 (stereoscopic environment)

Marín-Morales, J., Higuera-Trujillo, J. L., Greco, A., Guixeres, J., Llinares, C., Scilingo, E. P., ... & Valenza, G. (2018). Affective computing in virtual reality: emotion recognition from brain and heartbeat dynamics using wearable sensors. Scientific reports, 8(1), 1-15.

First study which uses EEG in an immersive scenario in combination with machine learning algorithm to recognize emotional states

Higuera-Trujillo, J. L., Maldonado, J. L. T., & Millán, C. L. (2017). Psychological and physiological human responses to simulated and real environments: A comparison between Photographs, 360 Panoramas, and Virtual Reality. Applied ergonomics, 65, 398-409.

It performs a comparision between 2D, 360º and 3D psychological and physiological responses also involving emotional domains. This suppose an important previos research in the comparision of 2D and VR content..

Marín-Morales, J., Higuera-Trujillo, J. L., Greco, A., Guixeres, J., Llinares, C., Gentili, C., ... & Valenza, G. (2019). Real vs. immersive-virtual emotional experience: Analysis of psycho-physiological patterns in a free exploration of an art museum. PloS one, 14(10), e0223881.

It performs a direct comparison between a real and VR environments, including EEG.

I think all those references are important contributions to the scope of the research stated in RQ1: Is there a physiological difference in emotional arousal stimulated from VR-2D and VR-3D? Therefore, the introduction need to be updated and explain better the current state of the art, where many improvements have been done in last years.

- The data processing techniques are very vague. How was managed the EDA artifacts? How EDA peaks are calculated? The artifacts correction need to be detailed in EEG considering the interactions of teh HMD and EEG. What are the threshold to reject participants? What do you do with bad channels? Many information is missed...

- Table 2 did not divide the self-assessment between 2d-3d conditions

- Tabke 3 did not show standard deviations.

- The discussion do not include any limitations derived from the methodology uses: number of stimuli, number of participants, possible artifacts derived from EEG and HMD...

- Brain synchronization analysis would be great to be performed (or at least discussed)

Minor:

- The first paragraph in chapter 2 seems more related to introduction than the materials and methods.

- "The self-rating scales for depression and anxiety were filled out before the experiment, and the scores were all within the normal range." What scales are used?

6. PLOS authors have the option to publish the peer review history of their article (what does this mean?). If published, this will include your full peer review and any attached files.

Reviewer #1: **Yes: **Javier Marín-Morales

---

## [Author Response · Author response to Decision Letter 0]

16 Jun 2021

Responses to Reviewers

Title: Emotional arousal in 2D versus 3D virtual reality environments

Journal: PLOS ONE

Author: Feng Tian, Minlei Hua, Wenrui Zhang, Yingjie Li*, Xiaoli Yang 

Comments to the Author

Thank you for submitting your manuscript to PLOS ONE. After careful consideration, we feel that it has merit but does not fully meet PLOS ONE’s publication criteria as it currently stands. Therefore, we invite you to submit a revised version of the manuscript that addresses the points raised during the review process.

1.Is the manuscript technically sound, and do the data support the conclusions?

Reviewer #1: Partly

R: Thank you for your suggestion. Our manuscript drew a conclusion through the analysis of subjective data and objective physiological data (e.g., EEG and EDA). The experiment was strictly carried out, and the mental conditions of the participants and the data quantities and qualities were properly controlled. The conclusion of this paper was also appropriately drawn based on the obtained data.

2.Has the statistical analysis been performed appropriately and rigorously?

Reviewer #1: No

R: Thank you for your suggestion. In this paper, SPSS software was used to perform statistical analysis and research on data by the use of ANOVA.

3. Have the authors made all data underlying the findings in their manuscript fully available?

Reviewer #1: No

R: Thank you for your suggestion. We have modified the data table to make the data more complete.

4. Is the manuscript presented in an intelligible fashion and written in standard English?

Reviewer #1: No

R: Thank you for your suggestion. We have invited a professional organization to edit the paper in terms of English usage.

5. Review Comments to the Author

Reviewer #1: The paper present a timely analysis of the differences between 2d and 3d VR. It presents interesting insight for the scientific community. However, I have major concerns in particular with previous research and data processing.

Major concerns:

(1)The introduction is quite limited and the state of the art have not been properly shown. Many references need to be considered

Marín-Morales, J., Llinares, C., Guixeres, J., & Alcañiz, M. (2020). Emotion recognition in immersive virtual reality: From statistics to affective computing. Sensors, 20(18), 5163.

Emotion recognition review using immersive VR, which summarize all the research performed in the emotion field with VR, in particular comparing 360º and 3D stimuli, which is quite related to RQ3 (stereoscopic environment)

Marín-Morales, J., Higuera-Trujillo, J. L., Greco, A., Guixeres, J., Llinares, C., Scilingo, E. P., ... & Valenza, G. (2018). Affective computing in virtual reality: emotion recognition from brain and heartbeat dynamics using wearable sensors. Scientific reports, 8(1), 1-15.

First study which uses EEG in an immersive scenario in combination with machine learning algorithm to recognize emotional states

Higuera-Trujillo, J. L., Maldonado, J. L. T., & Millán, C. L. (2017). Psychological and physiological human responses to simulated and real environments: A comparison between Photographs, 360 Panoramas, and Virtual Reality. Applied ergonomics, 65, 398-409.

It performs a comparision between 2D, 360º and 3D psychological and physiological responses also involving emotional domains. This suppose an important previos research in the comparision of 2D and VR content..

Marín-Morales, J., Higuera-Trujillo, J. L., Greco, A., Guixeres, J., Llinares, C., Gentili, C., ... & Valenza, G. (2019). Real vs. immersive-virtual emotional experience: Analysis of psycho-physiological patterns in a free exploration of an art museum. PloS one, 14(10), e0223881.

It performs a direct comparison between a real and VR environments, including EEG.

I think all those references are important contributions to the scope of the research stated in RQ1: Is there a physiological difference in emotional arousal stimulated from VR-2D and VR-3D? Therefore, the introduction need to be updated and explain better the current state of the art, where many improvements have been done in last years.

R: Thank you for your suggestion. We have carefully read the references provided by the reviewers and thought that they were very useful for our research, so they were all added to this paper. We appropriately supplemented the paper with the latest literature and introduced the relevant technical progress. We rewrote this part as follows:

“According to Marin-Morales' literature review[5], immersive VR, which allows researchers to simulate environments under controlled laboratory conditions with high levels of sense of presence and interactivity, is becoming more popular in emotion research. This work highlights the evolution of immersive VR use and in particular, the use of head-mounted displays, in emotion recognition research in combination with implicit measures.” (page 2)

“Higuera-Trujillo et al. [12] collected EDA and heart rate variability (HRV) data to compare 2D, 360° and VR. Their analysis showed that VR offers results that are closest to reality according to the physiological responses of the participants. These are important previous research efforts regarding comparisons of 2D and VR content. ”(page 3)

“IIn recent years, a combination of the characteristics of EEG data and machine learning algorithms to identify emotional states has become a new method. Marín-Morales et al.[21] performed an experiment that involved EEG and electrocardiography (ECG) recordings. They extracted features from these signals and fed them into a support vector machine (SVM) classifier to predict the arousal and valence perceptions of the subject. This is the first study that used EEG in an immersive scenario in combination with a machine learning algorithm to recognize emotional states. ”(page 3)

“ Later, in 2019, Marín-Morales et al. [22] developed an emotion recognition system using the SVM method combined with HRV and EEG data to identify arousal and valence. The literature showed that 3D is very valuable in applied research and analysis.” (page 3)

To better show the differences between VR-2D and VR-3D in this paper, we added Figure 1 to explain the research objects in this paper. (page 2)

“Fig 1. Comparison of internal helmet images in VR-2D and VR-3D modes.”

(2)The data processing techniques are very vague. How was managed the EDA artifacts? How EDA peaks are calculated? The artifacts correction need to be detailed in EEG considering the interactions of teh HMD and EEG. What are the threshold to reject participants? What do you do with bad channels? Many information is missed...

R: We thank the reviewer for pointing this out. We explained and supplemented your questions one by one in the article. First, we added the following regarding how EDA artifacts are managed and how EDA spikes are calculated, and the signal processing flowchart is also provided to help clarify this. “EDA data can be directly exported into MATLAB after data sampling. EDA data management and extraction used Ledalab, which is free software developed by Benedek et al. [26], for data optimization processing. In the data optimization process, data that are preset within the response window and meet the minimum amplitude criteria are determined to be peak SCRs that are caused by the stimuli. The flowchart for the signal processing used in this study is shown in Fig 6.” (page 5)

“Fig 6. The flowchart of EDA data processing.”

In the revision, the artifact removal and correction processes are supplemented, and the signal processing flowchart is also provided to help clarify these steps. "All collected EEG data were preprocessed in offline mode using the EEGLAB toolbox, including baseline adjustments and artifact removal from the EEGs and HMDs. The EEG signal processing flowchart for this study is shown in Fig 7.” (page 5)

“Fig 7. The flowchart of EEG data processing.”

To address bad channels, we have made the following additions: “For the problem of signal generation on some channels, we interpolated the data segment of the channel, that is, we replaced the signal value of the bad channel with data from 2-3 channels adjacent to the channel as a star interpolation.” (page 5)

Regarding the threshold for rejecting participants, we have added the following supplement to the article: “Data from total of twenty VR-2D and VR-3D subjects were collected. In the process of brain electrical signal preprocessing, the SDS scores of one subject in each of the VR-2D and VR-3D groups did not meet the experimental requirements. Therefore, the final data used in this experiment were from 19 people in the VR-2D group and 19 people in the VR-3D group.” (page 6)

(3)Table 2 did not divide the self-assessment between 2d-3d conditions.

R: We thank the reviewer for pointing this out. We have modified and improved the data presentation in Table 2 and divided the self-assessment into 2D and 3D conditions.

TABLE 2. The average value (mean ± S.D.) of valence and arousal in SAM (38 volunteers).

(4)Table 3 did not show standard deviations.

R: We thank the reviewer for pointing this out. We have modified and improved the display of data in Table 3 and have added the standard deviation data.

TABLE 3. The avgSCRs (mean ± S.D.) and numSCRs under different movie-watching modes.

(5)The discussion do not include any limitations derived from the methodology uses: number of stimuli, number of participants, possible artifacts derived from EEG and HMD.

R: We thank the reviewer for pointing this out. In our discussion, we added the limitations of the number of subjects and of the artifacts between EEG and HMD.

“The limitations of this paper include the following: (1) The number of subjects participating in the experiment is not sufficiently large, which limits our analysis of the data. In the future, if EEG information can be classified and recognized with machine learning, more subjects are needed, and more data are collected. (2) The artifacts between the HMD and EEG helmets easily effect data quality. To better handle data artifacts, the data should be processed in a more scientific way to eliminate the subjectivity of manual data processing.” (page 11)

In addition, in the discussion, we also added the relevant content from the newly cited literature to the discussion section, as follows: “ In Higuera-Trujillo's literature[12], the VR used for comparison is a VR-3D scene generated by a game engine, and 360° is the actual shot VR-2D mode. However, for this article, VR-3D and VR-2D were both obtained by shooting real scenes. Compared with game engine rendering, there are still visual differences in the details and realism of the pictures. Our conclusion also further confirms the results of Higuera-Trujillo, that is, VR-3D can produce greater emotional arousal, which may be caused by the stronger sense of reality of stereo vision.” (page 11)

(6)Brain synchronization analysis would be great to be performed (or at least discussed)

R: We agree. For brain synchronization analysis, our team has added relevant content from another paper that is related to the study of VR-2D and VR-3D. Therefore, we included a supplementary discussion of the relevant knowledge in this study as follows:

“When the brain is active, the neural activities of different regions are integrated at various spatial and temporal scales; this is termed the brain synchronization phenomenon. Li et al.[33] used the spectral features of the brain network to extract its synchronization features and proposed a method based on the fusion of traditional features and spectral features to achieve an adjustment of patient brain network synchronization abilities. By studying the ability of brain networks to synchronize, we can focus on describing the states and differences in the interactions between the two brain regions. Thus, in another recent study by our group[34], we compared neutral video in VR-2D and VR-3D through brain synchronization analysis. By constructing multiband functional brain networks, we observed higher global efficiency in β3 (21-30 Hz) networks in the VR-3D group. This strongly proves the conclusion of this paper.” (page 11)

(7)Minor-The first paragraph in chapter 2 seems more related to introduction than the materials and methods.

R: We agree. We moved this text to Section 2.4.1 PANAS and SAM, and now the first paragraph in chapter 2 is:

“Several methods exist to measure emotion arousal, including self-reports, EEG and EDA[23]. “ In this paper, the subjective scale Positive and Negative Affect Schedule (PANAS) combined with Self-Assessment Manikin (SAM) was selected, and EEG and dermatoelectrical EDA were selected for objective physiological data collection.” (page 4)

(8)."The self-rating scales for depression and anxiety were filled out before the experiment, and the scores were all within the normal range." What scales are used?

R: We agree. We rewrote this sentence as follows:

“The Self-rating Depression Scale (SDS) was completed before the experiment and included two items related to mental and emotional symptoms, eight items related to somatic disorders, two items related to psychomotor disorders, and eight items related to depressive psychological disorders.” (pages 4)

---

## [Decision Letter · Decision Letter 1]

3 Aug 2021

Emotional arousal in 2D versus 3D virtual reality environment

PONE-D-20-38463R1

Dear Dr. Li,

We’re pleased to inform you that your manuscript has been judged scientifically suitable for publication and will be formally accepted for publication once it meets all outstanding technical requirements.

Kind regards,

Christos Papadelis, Ph.D.

Academic Editor

PLOS ONE

Additional Editor Comments (optional):

Reviewers' comments:

Reviewer's Responses to Questions

**Comments to the Author**

1. If the authors have adequately addressed your comments raised in a previous round of review and you feel that this manuscript is now acceptable for publication, you may indicate that here to bypass the “Comments to the Author” section, enter your conflict of interest statement in the “Confidential to Editor” section, and submit your "Accept" recommendation.

Reviewer #1: All comments have been addressed

2. Is the manuscript technically sound, and do the data support the conclusions?

Reviewer #1: Yes

3. Has the statistical analysis been performed appropriately and rigorously? 

Reviewer #1: Yes

4. Have the authors made all data underlying the findings in their manuscript fully available?

Reviewer #1: No

5. Is the manuscript presented in an intelligible fashion and written in standard English?

Reviewer #1: Yes

6. Review Comments to the Author

Reviewer #1: The authors adress all my previous concerns. The quality of the paper has strongly improved. Congrats.

7. PLOS authors have the option to publish the peer review history of their article (what does this mean?). If published, this will include your full peer review and any attached files.

Reviewer #1: **Yes: **Javier Marín-Morales

---

## [Editor Report · Acceptance letter]

31 Aug 2021

PONE-D-20-38463R1 

Emotional arousal in 2D versus 3D virtual reality environments 

Dear Dr. Li:

I'm pleased to inform you that your manuscript has been deemed suitable for publication in PLOS ONE. Congratulations! Your manuscript is now with our production department. 

Kind regards, 

on behalf of

Dr. Christos Papadelis 

Academic Editor

PLOS ONE